# AuxiLight: Fast, Lightweight Auxiliary Loss Balancing Algorithm with Application in 6D Pose Estimation

## Abstract

Jointly learning multiple tasks has proven to be beneficial. Auxiliary task learning builds on this by jointly training a set of predefined auxiliary tasks to improve the performance of a desired main task. Unfortunately, auxiliary task learning is often not practical because 1) jointly training multiple tasks requires an exponential hyperparameter search space of task loss weights; 2) the auxiliary tasks often lead to expensive annotation costs to obtain ground truth. In this work, we propose *AuxiLight*, a generic auxiliary learning algorithm, and consider the concrete real-world use case of 6D pose estimation. AuxiLight addresses the first issue via an algorithm that adaptively balances the auxiliary task losses based on an analysis of the training dynamics. Experiments on standard multi-task datasets show that our method consistently outperforms single-task models and state-of-the-art auxiliary task learning methods, being the fastest and the most lightweight among the known task-weighting algorithms. To demonstrate the practicality of auxiliary learning on real-world tasks, we further apply our method to 6D object pose estimation. We highlight that, for this task, multiple ground-truth auxiliary annotations can in fact be generated for free. This lets us showcase a concrete use of auxiliary learning for real-world problems that does induce annotation costs.

## 1 Introduction

In the field of deep learning, jointly learning multiple tasks has long been observed to improve the performance of the model compared to one trained with a single task. Such practice, known as multi-task learning, has been used in all kinds of deep learning applications, such as computer vision, natural language processing, and drug discovery (Standley et al., 2020; Zhang & Yang, 2021; Eftekhar et al., 2021). Indeed, when different tasks are jointly learned together, the knowledge of one task can be often learned by another task, and the generalization ability of the deep model is expected to be improved. In light of this, the notion of *auxiliary-task learning* has been recently proposed (Zhang et al., 2014; Navon et al., 2020; Liu et al., 2022) as a special case of multi-task learning. The auxiliary tasks aim to improve the performance of a desired single task, referred to as the "main task", by jointly training all tasks together. Unlike multi-task learning where the performances of all tasks are evaluated, in auxiliary-task learning only the performance of the main task is considered; the auxiliary tasks are solely used to improve the performance of the main task.

Unfortunately, auxiliary-task learning is often not practical in improving the performance of real-world tasks. This is due to the following barriers:

- The contribution of different auxiliary tasks to the main task is unknown. For example, when the main task is surface normal estimation, one could imagine that the task of depth estimation contributes more than object detection (Standley et al., 2020). As a result, different auxiliary task losses should have different weights when jointly trained with the main task. However, doing so requires an exponential hyperparameter search space of task loss weights, which makes tuning the network impossible when multiple auxiliary tasks are present. While many works have addressed this issue (Navon et al., 2020; Standley et al., 2020; Du et al., 2018), via multiple algorithms to balance auxiliary task losses, we will see in Sec. 5 that they either have a bad performance, or are too computationally demanding to implement on any real-world tasks.

- Ground-truth annotations for auxiliary tasks often introduce expensive annotation costs. Indeed, when the purpose of the auxiliary tasks is solely improving the main task, the annotation cost of the auxiliary tasks might outweigh the performance gain they bring about, which makes auxiliary-task learning less practical. The only real-world application of auxiliary-task learning we are aware of (Zhang et al., 2014) relies on the auxiliary ground truths being manually labeled.

In this work we introduce *AuxiLight*, a generic, fast, lightweight auxiliary-task loss-balancing algorithm, and showcase its practicality with a real-world annotation cost-free use case on 6D pose estimation. Specifically, we address the aforementioned barriers in auxiliary-task learning as follows:

**Weight balancing algorithm.** To develop an algorithm that balances different auxiliary tasks when they are jointly trained with the main task, we make the assumption that the optimal weights of the auxiliary task losses should dynamically evolve with time to improve generalization during the training stage. As a result, we jointly optimize the weights of the auxiliary tasks throughout training, which results in a dynamical auxiliary task loss balancing algorithm that we refer to as *AuxiLight*. Such optimization on the auxiliary task weights is carried out by theoretically analyzing the training dynamics of the auxiliary tasks using the so-called technique of gradient flow approximation. Our results on several standard multi-task datasets show that AuxiLight improves over the single-task model and outperforms the state-of-the-art auxiliary task balancing algorithms (Liu et al., 2022; Du et al., 2018) in terms of speed and memory. In terms of accuracy, AuxiLight is comparable to the state-of-the-art algorithms and performs better in two of the three evaluated tasks.

**Real-world use case of auxiliary learning.** To showcase a real-world use case of auxiliary task learning without hefty annotation costs, we apply our AuxiLight algorithm to the task of 6D object pose estimation, one of the key tasks in computer vision with many real-world applications, including virtual reality, autonomous driving, and robotics. Formally, given a 3D object model (e.g., a 3D mesh) and an RGB image captured by a calibrated camera, this task aims to predict the 3D orientation and 3D position of the object relative to the camera. We highlight that, while the annotations required for the additional tasks in multi-task learning or auxiliary task learning for scene understanding are expensive to obtain, in 6D object pose estimation, they come virtually for free, thanks to the availability of the 3D object model. Specifically, we investigate the use of four different auxiliary tasks: Depth estimation, surface normal prediction, principal curvature estimation, and image reconstruction. The first three exploit the 3D object model to obtain ground-truth annotations, and the latter relies solely on the input image. To the best of our knowledge, this constitutes the first annotation cost-free application that takes advantage of auxiliary tasks. We showcase the benefits and generality of AuxiLight in 6D object pose estimation using two state-of-the-art 6D pose estimation algorithms, WDR-Net (Hu et al., 2021) and ZebraPose (Su et al., 2022). Our approach outperforms the single-task models, as well as the state-of-the-art auxiliary task learning algorithms. Note that the runtime during inference remains the same as with a single task because the auxiliary tasks are only needed during training. In other words, our approach improves the test-time accuracy of a 6D pose estimation network without requiring additional manual annotations and without affecting inference speed.

To this end, we summarize our contribution as follows:

- We have proposed *AuxiLight*, an algorithm that adaptively balances the auxiliary task losses based on a theoretical analysis of the training dynamics, which is the fastest and the most lightweight among the known task-weighting algorithms.

- While existing auxiliary task learning is virtually impractical in real-world due to the cost of annotating the auxiliary ground truths, we applied our AuxiLight on the task of 6D pose estimation, where ground-truth auxiliary annotations can be generated for free. To our best knowledge, this is the first use case of auxiliary-task learning that is practical in real-world tasks.

## 2 Related Work

### 2.1 Multi-task Learning

Multi-task learning aims at jointly learning multiple tasks, achieving provable information gain compared to learning single tasks separately when the tasks are related (Ben-David & Schuller, 2003). In the context of deep learning, multi-task learning is often carried out by constructing an encoder-decoder network where the encoder is shared across all tasks (Kokkinos, 2017). By jointly training different tasks, the network aims to learn a robust and universal representation, hence improving model generalization (Zhang & Yang, 2021). Many works have focused on the methodology of multi-task learning, and we refer the reader to (Standley et al., 2020) for a detailed discussion.

**Auxiliary task learning.** In context, our goal is not to achieve the best possible results in all tasks, but to improve only the main task. In the general multi-task learning literature, this is referred to as *auxiliary learning* (Navon et al., 2020; Du et al., 2018; Liu et al., 2022), which aims to exploit auxiliary tasks as implicit supervision to improve the performance of one single task. However, auxiliary learning does not come for free; training for the auxiliary tasks requires the corresponding ground truth, which, in general, involves an expensive annotation cost. For example, Zamir et al. (2018); Standley et al. (2020); Eftekhar et al. (2021) had to create new datasets with all tasks annotated to evaluate their proposed multi-task learning algorithm; in the context of facial landmark registration, Zhang et al. (2014) relied on manual annotations for auxiliary tasks such as gender recognition and predicting the presence of glasses. By contrast, in our constructed use case of 6D pose estimation, we leverage the fact that this specific task relies on the availability of a 3D object model, and propose auxiliary tasks whose ground truths can be automatically generated.

**Adaptive task balancing.** In the multi-task learning scenario, the task-specific losses are combined into an overall loss to supervise the network. The performance of the model is known to be sensitive to the weighting of each task-specific loss (Liu et al., 2022; Navon et al., 2020). Therefore, it is essential to carefully balance the loss of each task. The most naive way to do so is to perform a grid search on the task weights. However, the search space grows exponentially with the number of tasks, hence limiting the applicability of this strategy. Recent research has thus focused on automatically balancing the task weights (Liu et al., 2022; Navon et al., 2020; Du et al., 2018; Jiang et al., 2024; Mahapatra & Rajan, 2020; Lin et al., 2019). In particular, Du et al. (2018) heuristically use the cosine similarity of the task-specific loss gradients w.r.t. the shared encoder as task weights; Jiang et al. (2024) improve auxiliary task learning by mitigating a so-called negative transfer of auxiliary tasks; Auto-$\lambda$ (Liu et al., 2022) explicitly formalizes task loss weighting as a bi-level optimization problem, the solution of which can be obtained by calculating the second-order gradients. As a result, although achieving state-of-the-art results, Auto-$\lambda$ yields heavy computations and GPU memory usage, making the algorithm inapplicable to real-world scenarios, including 6D pose estimation. In this paper, we propose a fast, lightweight task loss balancing scheme that can be used for large models such as 6D pose estimation ones. Our method is derived from a theoretical analysis of the network dynamics and bears much less computational costs than Auto-$\lambda$ with comparable results.

### 2.2 6D Object Pose Estimation

In our constructed annotation cost-free auxiliary-task learning use case, we focus on 6D pose estimation from a single RGB image. Below, we discuss the methods that address the same task, either by direct prediction, or by exploiting a P$n$P-related step.

**Direct methods.** Early attempts (Kendall et al., 2015; Kehl et al., 2017; Xiang et al., 2017) aimed at directly regressing the 6D pose by training an end-to-end encoder-decoder CNN network. However, it is commonly believed that CNNs are ill-suited to directly map the input image into rotation and translation (Rad & Lepetit, 2017; Hu et al., 2019), and as a result, most state-of-the-art methods rather follow a geometry-inspired approach, predicting 3D-to-2D correspondences that can be exploited in P$n$P algorithm (Lepetit et al., 2009) to obtain the pose.

**P$n$P-based sparse methods.** A common strategy to obtain 3D-to-2D correspondences consists of predicting the image locations of a set of sparse object keypoints (Rad & Lepetit, 2017; Hu et al., 2019; Peng

et al., 2019). For example, BB8 (Rad & Lepetit, 2017) defines the keypoints as the eight 3D object bounding box corners. Hu et al. (2019) extend this idea by predicting the 2D keypoint locations for every pixel in the object mask. PVNet (Peng et al., 2019) predicts the direction from the object pixels to the 2D keypoints to establish the correspondences. However, as these methods only exploit a few keypoints, they tend to neglect the full geometry of the object model.

**P$n$P-based dense methods.** As a consequence, the recent state-of-the-art methods predict dense 3D-to-2D correspondences instead of the sparse ones (Di et al., 2021; Li et al., 2019; Su et al., 2022). This is achieved by predicting the corresponding 3D coordinate on the model surface for each object pixel in the input image. The state-of-the-art ZebraPose (Su et al., 2022) encodes the object mesh vertices into binary codes, turning 3D-to-2D correspondence prediction into a classification task instead of a 3D coordinate regression task.

Following this trend, we argue that exploiting the object model is important to improve the 6D pose estimation algorithms. In our work, however, we further exploit the object model by rendering several geometrical features in addition to the 3D-to-2D correspondence. Leveraging the idea of auxiliary learning discussed in Section 2.1, we use these model-induced features to implicitly supervise the network in an auxiliary task learning fashion.

## 3 AuxiLight: Balancing Auxiliary Tasks

Let us now introduce *AuxiLight*, our generic auxiliary task loss balancing algorithm. In Sec. 4, we will see a concrete use case of AuxiLight algorithm.

**Preliminiaries** Consider a general framework of auxiliary learning, where we have one main task $\mathcal{T}_0$ and a bunch of auxiliary tasks $\mathcal{T}_1, ..., \mathcal{T}_n$. The neural network is a multi-task hard parameter sharing network with a shared encoder $\boldsymbol{\theta}_{\mathrm{enc}}$ and different heads $\boldsymbol{\theta}_0; \boldsymbol{\theta}_1, ..., \boldsymbol{\theta}_n$, respectively for the main task and each auxiliary task. This is formalized as

$$\hat{y}_i = f_i(x; \boldsymbol{\theta}_{\mathrm{enc}}, \boldsymbol{\theta}_i), \quad i = 0, 1, ..., n,$$

where $x$ is the ground truth input and $\hat{y}_i$ is the predicted output for task $\mathcal{T}_i$. Additionally denote $y$ as the ground truth target. With the prediction and the ground target, each task $\mathcal{T}_i$ is associated with a loss, which is a function of the encoder and the corresponding head:

$$L_i(y_i, \hat{y}_i) = L_i(\boldsymbol{\theta}_{\mathrm{enc}}, \boldsymbol{\theta}_i).$$

### 3.1 Warm-up: Single-task learning

To better introduce the mechanism of auxiliary learning, we first take a look at the standard single-task learning (STL), where only the main task presents with no auxiliary task. Given training set $\{(x^{(k)}, y^{(k)})\}_{k=1,\cdots,K}$ sampled from an unknown (real) data distribution $D$, STL aims to minimize the expected loss

$$\bar{L}_0 = \mathbb{E}_{(x,y) \sim D} L_0(\boldsymbol{\theta}_{\mathrm{enc}}, \boldsymbol{\theta}_0). \tag{1}$$

As this expected loss is unknown, the population loss on the training set is minimized instead. Practically, people run SGD algorithm on a mini-batch loss, which follows the following formula:

$$\begin{aligned}
\boldsymbol{\theta}_{\mathrm{enc}} &\leftarrow \boldsymbol{\theta}_{\mathrm{enc}} - \alpha \cdot \frac{\partial L_0(\boldsymbol{\theta}_{\mathrm{enc}}, \boldsymbol{\theta}_0)}{\partial \boldsymbol{\theta}_{\mathrm{enc}}}, \\
\boldsymbol{\theta}_0 &\leftarrow \boldsymbol{\theta}_0 - \alpha \cdot \frac{\partial L_0(\boldsymbol{\theta}_{\mathrm{enc}}, \boldsymbol{\theta}_0)}{\partial \boldsymbol{\theta}_0}.
\end{aligned} \tag{2}$$

While SGD is hard to analyze theoretically, many recent papers have been investigating on gradient flow instead (Mei et al., 2019). Gradient flow is the continuous-time analogue to the discrete-time gradient step in Eq. 2:

$$\begin{aligned}
\frac{d\boldsymbol{\theta}_{\mathrm{enc}}(t)}{dt} &= -\alpha \cdot \frac{\partial L_0(\boldsymbol{\theta}_{\mathrm{enc}}, \boldsymbol{\theta}_0)}{\partial \boldsymbol{\theta}_{\mathrm{enc}}}, \\
\frac{d\boldsymbol{\theta}_0(t)}{dt} &= -\alpha \cdot \frac{\partial L_0(\boldsymbol{\theta}_{\mathrm{enc}}, \boldsymbol{\theta}_0)}{\partial \boldsymbol{\theta}_0}.
\end{aligned} \tag{3}$$

It has been shown that, with small learning rate $\alpha$, the gradient flow is a good approximation to the gradient step (Nesterov, 2003; Santambrogio, 2017; Scheithauer). With the same technique, our proposed auxiliary learning algorithm will be based on the gradient flow of a combined loss of the main task and the auxiliary tasks.

### 3.2 AuxiLight: Auxiliary learning

Same as STL, the main goal of auxiliary task is to minimize the expected loss of the main task (Eq. 1), but instead, auxiliary learning minimizes a surrogate multi-task loss

$$L = L_0(\boldsymbol{\theta}_{\mathrm{enc}}, \boldsymbol{\theta}_0) + \sum_{i \geq 1} \lambda_i L_i(\boldsymbol{\theta}_{\mathrm{enc}}, \boldsymbol{\theta}_i),$$

where $\lambda_i$ is the weight of the auxiliary task $\mathcal{T}_i$. Our proposed AuxiLight algorithm lies in the assumption that, $\lambda_i$ should be undetermined and evolve over time to improve generalization, i.e. $\lambda_i = \lambda_i(t)$ is function of time. As a result, we solve the best $\lambda$ by minimizing the expected main loss $\bar{L}_0$ as defined in Eq. 1. This is done by iteratively updating $\lambda_i$, with a gradient flow step as follows

$$\frac{d\lambda_i}{dt} = -\beta \cdot \frac{\partial \bar{L}_0}{\partial \lambda_i},$$

where $\beta$ is the learning rate for updating $\lambda$. Similar to Eq. 3 in STL, the training dynamics of the auxiliary-task network parameter is [1]

$$\frac{d\boldsymbol{\theta}_{\mathrm{enc}}}{dt} = -\alpha \cdot \frac{\partial L}{\partial \boldsymbol{\theta}_{\mathrm{enc}}} = -\alpha \sum_{i \geq 0} \lambda_i \cdot \frac{\partial L_i(\boldsymbol{\theta}_{\mathrm{enc}}, \boldsymbol{\theta}_i)}{\partial \boldsymbol{\theta}_{\mathrm{enc}}},$$

$$\frac{d\boldsymbol{\theta}_i}{dt} = -\alpha \cdot \frac{\partial L}{\partial \boldsymbol{\theta}_i} = -\alpha \lambda_i \cdot \frac{\partial L_i(\boldsymbol{\theta}_{\mathrm{enc}}, \boldsymbol{\theta}_i)}{\partial \boldsymbol{\theta}_i}.$$

Therefore by using chain rule, the dynamics of the task weighting can be calculated as follows: for $i \geq 1$,

$$\frac{d\lambda_i(t)}{dt} = -\beta \cdot \frac{\partial \bar{L}_0}{\partial \lambda_i(t)} = -\beta \cdot \frac{\partial \bar{L}_0(\boldsymbol{\theta}_{\mathrm{enc}}, \boldsymbol{\theta}_0)}{\partial \lambda_i(t)}$$

$$= -\beta \left( \left( \frac{\partial \boldsymbol{\theta}_{\mathrm{enc}}}{\partial \lambda_i(t)} \right)^\top \frac{\partial \bar{L}_0}{\partial \boldsymbol{\theta}_{\mathrm{enc}}} + \left( \frac{\partial \boldsymbol{\theta}_0}{\partial \lambda_i(t)} \right)^\top \frac{\partial \bar{L}_0}{\partial \boldsymbol{\theta}_0} \right).$$

It is worth noting that

$$\frac{\partial}{\partial \lambda_i(t)} \boldsymbol{\theta}_{\mathrm{enc}} = \frac{\partial}{\partial \lambda_i(t)} \left( \boldsymbol{\theta}_{\mathrm{enc}}^{(t=0)} + \int_{\tau=0}^t \frac{d\boldsymbol{\theta}_{\mathrm{enc}}}{dt} d\tau \right)$$

$$= -\alpha \frac{\partial}{\partial \lambda_i(t)} \int_{\tau=0}^t \sum_{i \geq 0} \lambda_i(\tau) \cdot \frac{\partial L_i(\boldsymbol{\theta}_{\mathrm{enc}}, \boldsymbol{\theta}_i)}{\partial \boldsymbol{\theta}_{\mathrm{enc}}} d\tau$$

$$= -\alpha \frac{\partial L_i(\boldsymbol{\theta}_{\mathrm{enc}}, \boldsymbol{\theta}_i)}{\partial \boldsymbol{\theta}_{\mathrm{enc}}} = -\alpha \frac{\partial L_i}{\partial \boldsymbol{\theta}_{\mathrm{enc}}},$$

where the derivative calculation in the last step has taken consideration of the fact that $\lambda(\tau)$ are independent of $\lambda(t)$ for all $\tau < t$. Following a similar calculation, we have $\partial \boldsymbol{\theta}_0 / \partial \lambda_i = \mathbf{0}$. Therefore, the weight of the auxiliary tasks follow the dynamics of

$$\frac{d\lambda_i(t)}{dt} = -\beta \left( \left( \frac{\partial \boldsymbol{\theta}_{\mathrm{enc}}}{\partial \lambda_i} \right)^\top \frac{\partial \bar{L}_0}{\partial \boldsymbol{\theta}_{\mathrm{enc}}} + \left( \frac{\partial \boldsymbol{\theta}_0}{\partial \lambda_i} \right)^\top \frac{\partial \bar{L}_0}{\partial \boldsymbol{\theta}_0} \right)$$

$$= \alpha\beta \cdot \left( \frac{\partial L_i}{\partial \boldsymbol{\theta}_{\mathrm{enc}}} \right)^\top \frac{\partial \bar{L}_0}{\partial \boldsymbol{\theta}_{\mathrm{enc}}}. \tag{4}$$

---

[1] For simplicity we denote fixed constant $\lambda_0 \equiv 1$.

In practice, the expected loss $\bar{L}_0$ is impossible to calculate. Following a similar argument to that of Liu et al. (2022), we use the current training batch instead to evaluate to gradient of the expected loss. By discretizing the gradient flow update of $\lambda_i$, we get the update as follows:

$$\lambda_i \leftarrow \lambda_i + \alpha\beta \cdot \left( \frac{\partial L_0}{\partial \boldsymbol{\theta}_{\text{enc}}} \right)^\top \frac{\partial L_i}{\partial \boldsymbol{\theta}_{\text{enc}}}. \tag{5}$$

The overall algorithm of our proposed AuxiLight algorithm is elaborated in Algorithm 1. Although the derivation seems strenuous, the update rule comes with an intuitive explanation. For each step, the task loss weight is increased by the inner product of the shared encoder gradient of the main task and the auxiliary task, which measures the similarity or relatedness between the auxiliary task and the main task. The auxiliary weight stops increasing at the later training stage when the two gradients become orthogonal, which means all knowledge of the auxiliary tasks have been learned.

---

**Algorithm 1 AuxiLight: Balancing Auxiliary Tasks**

---

**Input:** Main task $\mathcal{T}_0$, auxiliary tasks $\{\mathcal{T}_i\}_{i=1}^n$
**Input:** Corresponding loss functions $L_i$, $i \geq 0$
**Output:** Trained network of main task $\boldsymbol{\theta}_{\text{enc}}^*, \boldsymbol{\theta}_0^*$
  1: initialize learning rates $\alpha, \beta$
  2: initialize model parameters $\boldsymbol{\theta} = \{\boldsymbol{\theta}_{\text{enc}}, \{\boldsymbol{\theta}_i\}_{i=0}^n\}$
  3: initialize auxiliary task weights $\boldsymbol{\lambda} = [\lambda_1, \cdots, \lambda_n]$
  4: **while** NETWORK NOT CONVERGED **do**
  5:     $L \leftarrow L_0 + \sum_{i=1}^n \lambda_i \cdot L_i$
  6:     $\boldsymbol{\theta} \leftarrow \boldsymbol{\theta} - \alpha \cdot \nabla_{\boldsymbol{\theta}} L$
  7:     **for** $i = 1, \cdots, n$ **do**
  8:         $\lambda_i \leftarrow \lambda_i + \alpha\beta \cdot \nabla_{\boldsymbol{\theta}_{\text{enc}}} L_0^\top \nabla_{\boldsymbol{\theta}_{\text{enc}}} L_1$
  9:     **end for**
10: **end while**
11: **return** $\boldsymbol{\theta}_{\text{enc}}, \boldsymbol{\theta}_0$

---

In Sec. 5, we will see that AuxiLight is able to improve the performance of the single main task model. Compared to existing auxiliary loss balancing algorithms (Du et al., 2018; Liu et al., 2022), our algorithm requires the least runtime and GPU memory while achieving a comparable accuracy.

## 4 Annotation Cost-free Auxiliary Learning: A Case Study on 6D Pose Estimation

As discussed in Sec. 1, one of the barriers to practising auxiliary learning lies in the expensive annotation costs of the auxiliary tasks. In this section, however, we discuss a use case of auxiliary learning where annotations for auxiliary tasks can be obtained for free, with the main task being object 6D pose estimation. Our proposed auxiliary 6D pose estimation framework builds on any existing P$n$P-based 6D pose estimation algorithm (Hu et al., 2019; Su et al., 2022; Hu et al., 2021; Li et al., 2019; Di et al., 2021), and is empirically shown to be able to improve the performance of these existing methods with little annotation cost.

Below, we fist briefly present our multi-task model, then introduce the four auxiliary tasks we will study in our experiments, and finally show how auxiliary learning can be used to improve existing 6D pose estimation methods.

### 4.1 Main-task Model

The task of object 6D pose estimation is formally defined as follows. Given an object 3D model, represented by its $n$ surface vertices $\{\mathbf{p}_i \in \mathbb{R}^3\}_{i=1}^n$, object 6D pose estimation aims at predicting the rotation matrix $\mathbf{R} \in \mathbb{R}^{3\times3}$ and the translation vector $\mathbf{t} \in \mathbb{R}^3$ (i.e. 6D pose) of the object from an input RGB image $\mathbf{RGB} \in \mathbb{R}^{2\times H \times W}$. Additionally, the camera has a known intrinsic matrix $\mathbf{K} \in \mathbb{R}^{3\times3}$, and a 3D point $\mathbf{p} = [x, y, z]^\top$ on the 3D

model surface and its projected pixel $\mathbf{u} = [u, v]^\top$ in the image space should follow the perspective projection equation:

$$c[\mathbf{u}] \cdot \begin{bmatrix} u \\ v \\ 1 \end{bmatrix} = \mathbf{K} \left( \mathbf{R} \begin{bmatrix} x \\ y \\ z \end{bmatrix} + \mathbf{t} \right), \tag{6}$$

where $c[\mathbf{u}]$ is the depth at pixel $\mathbf{u}$.

As discussed in Sec. 2.2, instead of directly predicting the 6D pose $(\mathbf{R}, \mathbf{t})$, most existing methods predict the 3D-to-2D correspondence $\{\mathbf{p} \leftrightarrow \mathbf{u}\}$, followed by a P$n$P algorithm that solves 6D pose from the 3D-to-2D correspondence. Practically, such correspondence is predicted by regressing $\mathbf{u}$ for a fixed set of $\mathbf{p}$ for sparse methods, or by regressing $\mathbf{p}$ for every $\mathbf{u}$ visible in the image for dense methods. In any case, predicting 3D-to-2D correspondence is carried out by an encoder-decoder network and all existing methods can by summarized and formalized as follows:

$$\mathbf{pred}_{3D-2D}, \mathbf{pred}_{\text{mask}} = f_{\text{dec}}(\mathbf{repr}; \boldsymbol{\theta}_{\text{dec}}),$$

where

$$\mathbf{repr} = f_{\text{enc}}(\mathbf{RGB}; \boldsymbol{\theta}_{\text{enc}})$$

is the latent representation extracted by the encoder $f_{\text{enc}}$ with parameters $\theta_{\text{enc}}$, and processed by the decoder $f_{\text{dec}}$ with parameters $\theta_{\text{dec}}$ to obtain the 3D-to-2D correspondences. Normally an object mask $\mathbf{pred}_{\text{mask}}$ is predicted jointly for loss evaluation.

## 4.2 Multi-task Architecture

We first describe our multi-task model for auxiliary-task object 6D pose estimation. Our model is based on any existing 6D pose estimation algorithms that follow encoder-decoder architecture, as depicted in Sec 4.1. To jointly learn auxiliary tasks and improve prediction accuracy, we adopt a hard parameter sharing multi-task learning formalism (Kokkinos, 2017), where the latent representation $\mathbf{repr}$ is shared across all auxiliary tasks, and, for each task, a task-specific decoder is used to map the shared representation to the prediction. In other words, for the $i$-th task, we can write

$$\mathbf{pred}_{\text{task}_i} = f_{\text{dec}}(\mathbf{repr}; \boldsymbol{\theta}_{\text{task}-\text{dec}_i}),$$

where $\boldsymbol{\theta}_{\text{task}-\text{dec}_i}$ denotes the parameters of the decoder for the $i$-th task.

## 4.3 Proposed Auxiliary Tasks

Let us now define the auxiliary tasks that we will study in our experiments. As discussed in Sec. 1, the auxiliary tasks, in general, should (i) introduce minimum annotation costs with no manual labelling; (ii) contain shared knowledge with the main task that can be used to improve generalization. We highlight that, fortunately, for 6D pose estimation, many tasks meet the aforementioned requirements. Specifically, these tasks include predicting depth, surface normal, and principle curvature.

The details of the proposed auxiliary tasks are explained in the following paragraphs. Due to the fact that both ground-truth object 3D model and 6D pose are available, the scene can be perfectly reconstructed by using the perspective projection function Eq. 6. As a result, the ground truths of the auxiliary tasks can be generated effortlessly.

**Depth.** Depth plays an important role in 6D pose estimation, as it is directly related to the translation component of the 6D pose. This has been exploited by RGB-D 6D pose estimation methods, which take measured depth (He et al., 2020; Wang et al., 2019; Li et al., 2018) as input. Here, however, we do not take depth as input, but leverage the fact that it shares information with the 6D pose to exploit depth prediction as an auxiliary task. Formally, we can thus define the ground truth for our *depth task* as

$$\text{gt}_{\text{depth}}[\mathbf{u}] = c[\mathbf{u}]. \tag{7}$$

While this information is valuable when predicting 6D pose from the entire input image, it is ill-suited in the case where the object is first detected and cropped prior to pose estimation (Li et al., 2019; Su et al., 2022; Di et al., 2021), as absolute depth is hard to infer from a crop. To handle this scenario, for each pixel in the object mask, we calculate the distance in depth to the pixel with the smallest depth in the mask. This lets us define the ground truth for our *differential depth task* as

$$\text{gt}_{\Delta\text{depth}}[\mathbf{u}] = c[\mathbf{u}] - \min_{\mathbf{M}[\mathbf{u}']=1} c[\mathbf{u}'], \tag{8}$$

where $\mathbf{M}$ is the 0-1 object mask, with $\mathbf{M}[\mathbf{u}'] = 1$ if and only if pixel $\mathbf{u}'$ belongs to the object. Differential depth is invariant under cropping or zoom-in transformations.

**Surface normal.** Surface normal characterizes the first-order derivative of the object 3D shape, and is crucial for scene understanding. Multiple works have identified surface normal as a beneficial auxiliary task (Standley et al., 2020; Liu et al., 2022). Given the object 3D model, we can offline calculate the unit surface normal $\mathbf{n}$, $(\|\mathbf{n}\|_2 = 1)$ in the object space for every point $\mathbf{p}$ on the object surface. The surface normal is then rotated to the camera space according to the object pose to make the task predictable from a standalone image. This ground-truth information for the *surface normal task* can thus be calculated as

$$\text{gt}_{\text{normal}}[\mathbf{u}] = \mathbf{R} \cdot \mathbf{n}[\mathbf{p}]. \tag{9}$$

**Principal curvature.** Principal curvature captures the second-order properties of the object 3D shape. It is an excellent candidate for an auxiliary task because curvature is an intrinsic 3D shape property which is invariant under rotation and translation. When the 3D object model is provided as a triangulated mesh, the principal curvatures $\mathbf{k}_1, \mathbf{k}_2$ can be calculated using a discrete curvature operator (Cohen-Steiner & Morvan, 2003; Meyer et al., 2003). We denote the ground truth for the *principal curvature task* as

$$\text{gt}_{\text{curvature}}[\mathbf{u}] = \big[\mathbf{k}_1[\mathbf{p}], \mathbf{k}_2[\mathbf{p}]\big]. \tag{10}$$

**Image reconstruction.** Finally, we also include image reconstruction as an additional auxiliary task. This is also referred to as auto-encoding (Hinton & Salakhutdinov, 2006), with the encoder-decoder network being trained to produce the input image as output. While reconstruction does not capture the properties of the 3D object model, it can help to find a low-dimensional, robust-to-noise representation for the input image, potentially improving model generalization. The *reconstruction task* is thus formalized as

$$\text{gt}_{\text{reconst.}}[\mathbf{u}] = \mathbf{RGB}[\mathbf{u}], \tag{11}$$

where $\mathbf{RGB}[\mathbf{u}]$ is the RGB vector of the input image at pixel $\mathbf{u}$.

### 4.4 Auxiliary Task Training

To exploit the object 3D geometry and improve generalization from the auxiliary tasks, we train the multi-task model under the supervision of the auxiliary tasks defined in Eq. 7 - Eq. 11. Formally, we define the set of auxiliary tasks $\mathcal{S}$ as $\mathcal{S} = \{\text{depth}, \text{normal}, \text{curvature}, \text{reconst.}\}$ when the network input is the full image, and $\mathcal{S} = \{\Delta\text{depth}, \text{normal}, \text{curvature}, \text{reconst.}\}$ when the image is first cropped using a detector. Following Standley et al. (2020), we adopt the $\ell_1$ loss to supervise each task. Moreover, due to the fact that all tasks are only defined on the pixels of the visible part of the target object, the corresponding losses are confined to the object foreground as well. As a result, the loss for each $\text{task}_i \in \mathcal{S}$ can be formalized as

$$\mathcal{L}_{\text{task}_i} = \|\mathbf{M} \odot \mathbf{gt}_{\text{task}_i} - \mathbf{M} \odot \mathbf{pred}_{\text{task}_i}\|_1, \tag{12}$$

where $\mathbf{M} = \mathbf{gt}_{\text{mask}}$ is the 0-1 mask of the object, and $\mathbf{gt}$, $\mathbf{pred}$ are the ground truth and prediction of the corresponding task defined in Sec. 4.3.

With all the auxiliary tasks and their losses defined, now we can follow the AuxiLight algorithm in Sec. 3 to train the network. The overall training loss is defined as a linear combination of main loss and task-specific

| Algorithm | GPU Memory | # FW Pass | # BW Pass |
|---|---|---|---|
| No Auxiliary | $2M_{\text{enc}} + 2M_{\text{dec}}$ | 1 | 1 |
| Equal weight | $2M_{\text{enc}} + 2(|\mathcal{S}| + 1) \cdot M_{\text{dec}}$ | 1 | 1 |
| GradSim | $3M_{\text{enc}} + 2(|\mathcal{S}| + 1) \cdot M_{\text{dec}}$ | 1 | 1 |
| Auto-$\lambda$ | $5M_{\text{enc}} + 5(|\mathcal{S}| + 1) \cdot M_{\text{dec}}$ | 2 | 3 |
| **AuxiLight** | $3M_{\text{enc}} + 2(|\mathcal{S}| + 1) \cdot M_{\text{dec}}$ | 1 | 1 |

Table 1: **Memory & Runtime comparison between AuxiLight and other auxiliary task loss balancing algorithms.** We perform a theoretical analysis of the memory consumption and run time, where $M_{\text{enc}}$ denotes the memory for storing the shared encoder weights, $M_{\text{dec}}$ the memory for storing the decoder for each auxiliary task or the main task (assuming the same size), and $\mathcal{S}$ is the set of auxiliary tasks. # FW Pass and # BW Pass refer to the number of forward and backward passes needed with pyTorch implementation.

| | NYUv2 dataset: (a) Performance of main task ... | | | CityScapes dataset: (b) Performance of main task ... | | |
|---|---|---|---|---|---|---|
| Algorithm | Sem. Seg. mIOU ($\uparrow$) | Depth aErr. ($\downarrow$) | Normal mDist.($\downarrow$) | Sem. Seg. mIOU ($\uparrow$) | Pt. Seg. mIOU ($\uparrow$) | Disp. mDist. ($\downarrow$) |
| No Auxiliary | 44.31 | 48.41 | 22.96 | 56.20 | 52.74 | 0.84 |
| Equal weight | 46.87 | 41.13 | 23.63 | 55.72 | 52.62 | 0.83 |
| GradSim | 45.17 | 42.51 | 23.94 | 55.76 | 52.19 | 0.80 |
| Auto-$\lambda$ | 47.22 | 40.38 | **23.12** | 57.89 | 53.56 | **0.77** |
| **AuxiLight** | **47.29** | **40.24** | 23.40 | **58.50** | **53.58** | 0.80 |

Table 2: **Performance comparison between AuxiLight and other auxiliary task loss balancing algorithms.** We show the performance of the algorithms on each corresponding main task of (a) NYU-v2 and (b) CityScapes. The two tasks other than the main task are used as auxiliary tasks. Given limited computational resources, our method achieves the best performance and is even comparable with the more expensive state-of-the-art Auto-$\lambda$ method.

auxiliary losses, i.e.,

$$\mathcal{L} = \mathcal{L}_{\text{main}} + \sum_{i=1}^{|\mathcal{S}|} \lambda_{\text{task}_i} \cdot \mathcal{L}_{\text{task}_i}, \tag{13}$$

where $\lambda_{\text{task}_i}$ is the weight of the $i$-th auxiliary task $\text{task}_i \in \mathcal{S}$, and $\mathcal{L}_{\text{main}}$ is the original loss of the 6D pose estimation method, i.e., typically a loss combining the 3D-to-2D correspondence error and the segmentation error for predicting the object mask. The AuxiLight algorithm is then followed to train the auxiliary tasks and the main task jointly. We comment that, during test time, all auxiliary tasks are discarded, and the inference speed thus remains the same as the base model trained with the single main task.

## 5 Experiments

In this section, we first evaluate our AuxiLight algorithm on standard multi-task datasets and compare it with existing task loss balancing algorithms, then we report the results on auxiliary-task 6D pose estimation depicted in Sec. 4. We will make the code publicly available.

### 5.1 AuxiLight vs Other Task Balancing Schemes

We evaluate our AuxiLight algorithm on the NYUv2 dataset (Silberman et al., 2012) and the CityScapes dataset (Cordts et al., 2016), which are standard datasets widely used to evaluate multi-task algorithms. Specifically, for the NYUv2 dataset, we train for three tasks: semantic segmentation, depth estimation and

surface normal prediction; while on the CityScapes dataset, we train for semantic segmentation, disparity estimation and part segmentation. Following existing auxiliary weight balancing algorithms (Du et al., 2018; Liu et al., 2022), we carry out three experiments where one task is taken as the main task and the other two are the auxiliary tasks. We use the same multi-task learning framework with hard parameter sharing discussed in Section 4.2 as network architecture.

Furthermore, we compare our proposed AuxiLight with equal weights, GradSim (Du et al., 2018) and Auto-$\lambda$ (Liu et al., 2022). We aim to show that 1) Auto-$\lambda$, the state-of-the-art auxiliary task loss balancing algorithm, requires significant computational resources, making it ill-suited to large models; 2) our proposed AuxiLight is faster and more lightweight with comparable performance to Auto-$\lambda$.

**Memory & Runtime.** In Table 1 we show a memory and run-time comparison of all auxiliary task loss balancing algorithms. $M_{\text{enc}}$ denotes the memory for storing the shared encoder weights, $M_{\text{dec}}$ the memory for storing the decoder for each auxiliary task or the main task (without loss of generality, we assume that all decoders have the same size), and $\mathcal{S}$ is the set of auxiliary tasks. Note that in most neural network implementations (*e.g.*, pyTorch), storing the gradients requires an equal amount of memory as storing the parameters. To our surprise, Auto-$\lambda$ requires significantly more memory and computations than our proposed AuxiLight and other task loss balancing schemes. This is due to the fact that Auto-$\lambda$ involves calculating a second-order hessian of the weights, whose approximation requires at least twice as much memory and number of forward / backward passes. In fact, for our specific application of 6D pose estimation, the Auto-$\lambda$ algorithm is not able to run on a single GPU, which justifiably shows the drawback of this state-of-the-art algorithm.

**Accuracy.** In Table 2 we report the performance for the main task when the other two tasks are jointly trained as auxiliary task. The evaluation metrics are identical with those of Liu et al. (2022) and the details are elaborated in the appendix. Our proposed AuxiLight ranks first in two out of three cases, being second to only Auto-$\lambda$ in normal estimation. Importantly, AuxiLight consistently outperforms GradSim, sometimes by a large margin, making it the best option when computational resources are limited.

### 5.2 Auxiliary-task 6D Pose Estimation

We now introduce our experiments on auxiliary-task 6D pose estimation discussed in Sec. 4.

**Base models.** Our proposed multi-task 6D pose estimation framework can be applied to any existing P$n$P-based method that regresses the 6D pose from 3D-to-2D correspondences. To showcase this, following the taxonomy in Section 2.2, we choose one state-of-the-art 6D pose estimation method from each category, i.e., one P$n$P-based sparse method and one P$n$P-based dense method. Specifically, we use WDR-Net (Hu et al., 2021) and ZebraPose (Su et al., 2022) as base models in our experiments. WDR-Net establishes sparse 3D-to-2D correspondences by predicting 8 projected 2D keypoint coordinates using an FPN-based network. By contrast, ZebraPose predicts binary codes encoding 3D coordinates on the object surface to obtain dense 3D-to-2D correspondences, using a decoder adapted from Deeplabv3 (Chen et al., 2017). In both methods, a semantic object mask is jointly predicted to confine the 3D-to-2D correspondences within the object foreground.

**Multi-task framework.** To exploit our multi-task framework to improve existing 6D pose estimation algorithms, we first render the ground truths of the auxiliary tasks defined in Section 4.3. Given the object 3D mesh and the ground-truth 6D pose, the scene is rendered using OSMesa, an off-screen renderer, and the ground truths of task {depth, $\Delta$depth, normal, curvature} are consequently calculated according to Eq. 7 - Eq. 10. Note that generating the auxiliary ground truths is performed offline, automatically, and requires no expensive annotation costs.

The multi-task architecture follows Sec. 4.2. We use the same encoder as the base model and, for each auxiliary task, a task-specific decoder with the same decoder architecture as the base model, except for the last convolutional layer which is modified to match the number of channels of the target tasks.

| | (a) WDR-Net | | | | (b) ZebraPose | | | |
|---|---|---|---|---|---|---|---|---|
| Class | BM | +EQ | +GS | +AL | BM | +EQ | +GS | +AL |
| ape | 33.8 | 35.0 | 39.4 | 35.2 | 58.2 | 59.4 | 59.6 | 59.2 |
| can | 58.0 | 60.5 | 59.6 | 60.2 | 96.0 | 96.3 | 96.4 | 96.3 |
| cat | 48.7 | 50.3 | 47.6 | 50.7 | 61.0 | 60.0 | 60.7 | 60.8 |
| drill. | 63.9 | 65.5 | 65.2 | 65.0 | 95.0 | 94.5 | 93.9 | 93.7 |
| duck | 38.1 | 39.2 | 39.6 | 40.0 | 58.8 | 62.2 | 63.7 | 62.9 |
| egg. | 71.0 | 71.3 | 72.0 | 71.8 | 70.7 | 71.9 | 72.5 | 72.7 |
| glue | 67.5 | 68.2 | 67.3 | 67.0 | 89.4 | 90.1 | 90.2 | 90.2 |
| hol. | 47.0 | 48.5 | 46.9 | 50.5 | 81.0 | 81.0 | 81.7 | 80.1 |
| Avg. | 53.5 | 54.8 | 54.7 | **55.1** | 76.2 | 76.9 | **77.3** | 77.1 |

Table 3: **Results of our proposed multi-task object 6D pose framework.** We report the ADD-0.1d accuracy for each object class and for each task weighting scheme. BM is the original base model, either (a) WDR-Net or (b) ZebraPose, without auxiliary tasks. Three auxiliary task weighting schemes are compared here: EQ=equal Weights, GS=GradSim, AL=AuxiLight. As discussed in Sec. 5.2, the Auto-$\lambda$ algorithm is excluded from comparison because of its impracticability on large, real-world models.

In our experiments, we will compare three task balancing algorithms to optimize the multi-task model: 1) equal weights, 2) GradSim and 3) our proposed AuxiLight (Sec. 3). We excluded Auto-$\lambda$ from this set of experiments because of its too large memory and computation overhead for the task of 6D pose estimation, as discussed in Section 5.1.

**Datasets.** We perform our experiments on the 8-object Occ-LINEMOD dataset (Brachmann et al., 2014), which is the dataset used by both ZebraPose and WDR-Net, i.e., the two base models selected in this paper.

**Evaluation Metric.** The metric is the same as our selected base models. Specifically, we use the ADD(-S) metric to evaluate 6D pose estimation performance, which measures the point-to-point distance in 3D space between the 3D object vertices transformed by the predicted pose and the ground-truth pose. We report the ADD-0.1d accuracy, i.e. the percentage of samples for which the ADD(-S) error is smaller than 10% of the object diameter.

**Results on WDR-Net.** In Table 3 (a), we report the results of our proposed multi-task 6D pose model based on WDR-Net (Hu et al., 2021). These results evidence the benefits of the auxiliary tasks; all three task weighting schemes improve the performance of the base model. Nevertheless, our AuxiLight yields the largest boost, with an increased ADD-0.1d accuracy of 1.8 percent.

**Results on ZebraPose.** In addition to WDR-Net, we also test our multi-task method with ZebraPose as source model. The result are shown in Table 3 (b). Due to computational restrictions, we were not able to include all auxiliary tasks in our experiments, and only surface normal and reconstruction are included as auxiliary tasks. These tasks correspond to the individual tasks bringing the largest accuracy boost, as will be elaborated in Appendix. While here, GradSim has a slight advantage over AuxiLight, both task weighting schemes outperform the single-model baseline. Given the margin over GradSim on other cases, we still consider that AuxiLight is in general advantageous over GradSim.

# 6 Conclusion

In this work, we have introduced AuxiLight, a fast, lightweight auxiliary task weighting scheme that achieves comparable performance as the state-of-the-art algorithm, but at a much lower computational and memory burden. In addition, we have constructed an annotation cost-free use case of auxiliary learning based on 6D pose estimation methods, with the advantage that the tasks are freely-available via the 3D object models. Such annotation cost-free use case is the first of its kind to our best knowledge.

**Limitations.** As a case study, the auxiliary tasks used for 6D pose estimation do not automatically generalize to other tasks. Nevertheless, the general idea behind cost-free annotation is the fact that access to

a 3D model provides means to generate the ground truth for different tasks. In principle, this idea could be leveraged in other contexts, such as 3D reconstruction and 6D object tracking from images, or potentially 3D object part segmentation from point clouds, where auxiliary tasks could include curvature and signed distance prediction. Demonstrating this, however, goes beyond the scope of this work, but provides interesting directions for future research.

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

# A    Appendix

## A.1    Illustration of Auxiliary-Task Learning

To help readers better understand the scheme of the proposed auxiliary-task 6D pose estimation depicted in Sec. 4, we illustrate the framework in Fig. 1. Our proposed auxiliary-task learning framework (colored in red) can be employed to improve the training of existing 6D pose estimation methods (colored in grey). Solid arrows indicate the forward data flow, while dashed arrows show the backward loss flow. It is self-evident that 1) The ground truths of auxiliary tasks can be generated automatically with little annotation cost, thanks to the availability of 3D model in the specific task of 6D pose estimation; 2) Our auxiliary-task framework can be seen as an add-on that is not involved during inference, and thus yields a more effective model at no additional cost in test-time speed.

## A.2    Implementation Details of Sec. 5.1

In this section we elaborate on implementation details of the AuxiLight experiments that are carried out on the NYUv2 dataset (Silberman et al., 2012) and the CityScapes dataset (Cordts et al., 2016), which are omitted in Sec. 5.1. Our experimental setup, as well as code implementation largely follows (Liu et al., 2022), which is the state-of-the-art auxiliary loss balancing algorithm to date. We will release our code upon acceptance.

**Evaluated tasks.**    For the NYUv2 dataset, we train semantic segmentation, depth estimation and surface normal prediction; and for the CityScapes dataset, we train semantic segmentation, disparity estimation and part segmentation. The tasks of segmentation, depth and normal are respectively evaluated by mean intersection over union (IoU, the larger the better), absolute error (Err., the smaller the better), and mean angle distance error (Err., the smaller the better). For the task of disparity estimation on the CityScapes dataset, we calculate the depth error as well, as essentially disparity is the inverse of depth. For both datasets, we train three models where one task is the main task and the other two tasks are auxiliary tasks, and we

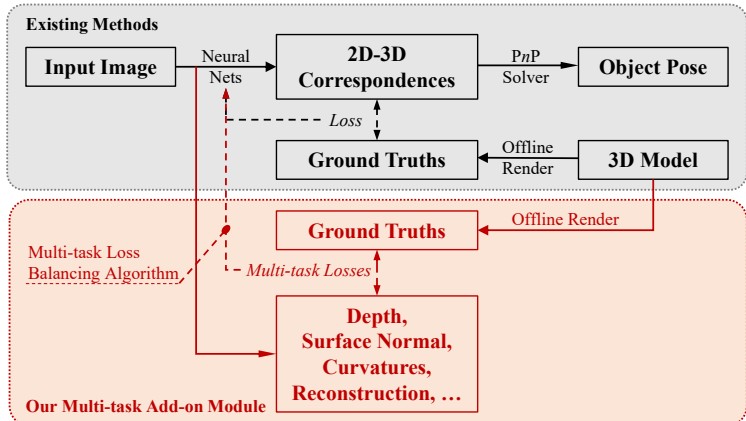

Figure 1: High-level summary of our proposed auxiliary-task learning framework (colored in red). Our framework can be employed to improve the training of existing 6D pose estimation methods (colored in grey). Solid arrows indicate the forward data flow, while dashed arrows show the backward loss flow. (Better viewed in color.)

| WDRNet | +depth | +normal | +curvature | +reconst. |
|--------|--------|---------|------------|-----------|
| 53.5   | 54.1   | 54.5    | 54.0       | 54.4      |
| ZebraPose | +$\Delta$depth | +normal | +curvature | +reconst. |
| 76.2   | 76.9   | 77.1    | 76.1       | 77.4      |

Table 4: **Result of our multi-task framework when one single auxiliary task is used.**

evaluate the performance of the main task with its corresponding metric. Following Liu et al. (2022), to make a fair comparison we add noise prediction as a third auxiliary task. Each model is trained and evaluated twice to reduce randomness.

### A.3 Additional Analysis of Sec. 4.4

In this section, we provide additional analysis for the auxiliary-task estimation depicted in Sec. 4.4.

**Memory analysis.** We compare the memory of our proposed AuxiLight and other auxiliary weight-balancing algorithms as follows. On the NYUv2 dataset, Equal weight uses 4944 MB memory, Auto-$\lambda$ uses 7948 MB memory, and our proposed AuxiLight uses uses 5154 MB memory. On the Cityscapes dataset, Equal weight uses 4206 MB memory, Auto-$\lambda$ uses 6002 MB memory, and our proposed AuxiLight uses 4352 MB memory. This confirms that our method uses much less memory than the state-of-the-art Auto-$\lambda$.

**Runtime analysis.** Reporting the actual runtime is more nuanced. On our cluster, every run is assigned with a fixed amount of GPU, but the CPUs are shared across all jobs. The runtime is related the current CPU load, which makes it hard for benchmarking. We nonetheless report the runtimes across all runs as follows: Equal weight takes 0.76s per step, Auto-$\lambda$ takes 0.92s per step, and our method takes 0.67s per step.

### A.4 Additional Analysis of Sec. 5.2

In this section, we provide additional analysis for the auxiliary-task 6D pose estimation depicted in Sec. 5.2.

**Contribution of the individual auxiliary tasks.** In addition to the previous experiments where multiple auxiliary tasks were used, we conduct experiments using one single auxiliary task at a time, to compare the contribution of the different auxiliary tasks. In this case, a fixed auxiliary weight is used. The results are

| Task | | | Average time per frame |
|---|---|---|---|
| Rendering Auxiliary GT of... | depth | | 240 ms |
| | normal | | 50 ms |
| | curvature | | 140 ms |
| | overall | | 430 ms |
| Training | | Baseline | 100 ms |
| | ZebraPose | + GradSim | 180 ms |
| | | + Auto-$\lambda$ | N/A |
| | | + AuxiLight | 180 ms |
| | | Baseline | 110 ms |
| | WDRNet | + GradSim | 140 ms |
| | | + Auto-$\lambda$ | N/A |
| | | + AuxiLight | 140 ms |
| Testing | | Baseline | 220 ms |
| | ZebraPose | + GradSim | 220 ms |
| | | + Auto-$\lambda$ | N/A |
| | | + AuxiLight | 220 ms |
| | | Baseline | 70 ms |
| | WDRNet | + GradSim | 70 ms |
| | | + Auto-$\lambda$ | N/A |
| | | + AuxiLight | 70 ms |

Table 5: **Runtime Analysis of Auxiliary-task 6D Pose Estimation.** The times are rounded up to multiples of 10ms. Results with Auto-$\lambda$ are simply marked with N/A because this algorithm requires too much GPU memory and running time that it fails with our experiment set-up (as discussed in Sec. 5.1).

shown in Table 4, and we show that, except for curvature with ZebraPose, all auxiliary tasks help improve the base 6D pose estimation model. Moreover, in both cases, surface normal and reconstruction rank as the top two auxiliary tasks.

**Runtime analysis.** Table 5 shows the running time of our proposed auxiliary-task 6D pose framework. Compared to the base model, our auxiliary-task framework entails two additional computations: 1) Generating auxiliary task ground truth annotations, and 2) training additional auxiliary branches. While rendering ground truths costs time, it is performed only once in an offline fashion. We highlight that 1) our proposed Auxi6D is indeed a fast, lightweight auxiliary loss balancing algorithm compared to other equivalents; and 2) our method does not introduce additional computation during inference; the additional auxiliary branches can be discarded.

