# OpenReview forum: "AuxiLight: Fast, Lightweight Auxiliary Loss Balancing Algorithm with Application in 6D Pose Estimation"
_TMLR — Rejected by TMLR_

### Review · Reviewer_DDnz · 2024-09-24

**Summary Of Contributions:**

The paper proposes a new method for auxiliary learning. The main idea is to adapt the weights of the auxiliary loss terms with a gradient-based method. The experiments in the paper suggest that AuxiLight performs on par or better to existing methods, while it can reduce runtime and memory costs.

**Audience:**

Yes

**Broader Impact Concerns:**

None.

**Claims And Evidence:**

No

**Requested Changes:**

See the weaknesses/questions above.

**Strengths And Weaknesses:**

Strengths:

The main idea and objective of the paper is clearly formulated: we want to design a method that can adaptively set weights of the auxiliary loss functions.
Several experiments are conducted for real-world datasets and with rather recent benchmark methods.

Weaknesses:

1) **Derivation of the method:** Regarding the derivation of the main method in section 3.2, I have the following question (writing $\theta$ for $\theta_{enc}$ for simplicity): on page 5, in the equation

\begin{equation}\partial \theta / \partial \lambda_i = [..] \end{equation}

it should be noted that the derivative $d\theta/dt$ inside the integral is actually the time-derivative evaluated at $\tau$.
Now, in the next step, we get that the term $\partial L_i / \partial \theta $ must also be evaluated at $\tau$.

Now, if we're allowed to swap derivative and integral (are we? if yes, please add more justification), then as the sum only contains one linear term in $\lambda_i$, we would get

\begin{equation}\int_{\tau=0}^t \frac{\partial L_i(\theta,\tau)}{\partial \theta} d\tau. \end{equation}

I don't think that this is the same as $\frac{\partial L_i(\theta,t)}{\partial \theta}$ in general.
Another minor comment: in the sum, it should be over $j\geq 0$ and not $i\geq0$, as we already picked a fixed $\lambda_i$ before.

Maybe I have misunderstood your derivation here, and in that case please correct me. However, in the other case, I am not sure about the motivation for the method. The above calculation suggests that you would need to integrate your gradients over the iteration trajectory, which is different from Algorithm 1.

2) Experiments

I am not an expert on Pose Estimation Problems, so my assessment of the experimental part should be taken into consideration attenuated only. One thing I observed is that the metrics for the NYUv2 run for the main baseline, Auto-lambda, are different in your submission to the original Auto-lambda paper. It looks to me that the experimental setup, and the reported metrics are (almost) identical to the Auto-lambda paper, so I wonder where the difference comes from.

For example, you report a Sem. Seg. oIOU of **47.22**, while Auto-lambda reports **48.04** for Split-3 tasks and **47.80** Split-1task. Both scores are better than the value you report AuxiLight. For CityScapes the values seem to match.

Another question on the experimental results is, if you can report actual memory and runtime usage of your runs? Table 1 only shows theoretical memory usage if I understood correctly, but it would be interesting to see whether this difference translates into actual consumption.


*I have marked Claims and Evidence as No mostly due to the concerns on the derivation of the method, formulated above. However, in case this can be clarified in the revision/discussion, I am happy to change my assessment.*

---

> ### Author Response · Authors · 2024-11-18
>
> We thank reviewer DDnz for acknowledging our contributions on real-world machine leaning tasks. Here are the detailed responses to the reviewer's questions:
>
> **1. Derivation of the method.**
>
> > ... Now, if we're allowed to swap derivative and integral (are we? if yes, please add more justification), as the sum only contains one linear term in $\lambda_i$, we would get ...
>
> Unfortunately, your proposed derivation is not correct, as the derivative and the integral are simply not exchangeable. We kindly invite the reviewer to look again at the calculation concerned: At time $t$,
> $$\frac{d\theta}{d\lambda_i} = -\alpha \frac{\partial}{\partial \lambda_i}\int_{\tau=0}^t \sum_{i \ge 0} \lambda_i \frac{\partial L_i(\theta, \tau )}{\partial \theta} d\tau.$$
> In the above function, the derivative operation $\partial / \partial \lambda_i$ is w.r.t. $\lambda_i$ at a specific time $t$, whereas $t$ is also the upper limit of the integral on the RHS, making the two operations nonexchangeable. Moreover, inside the integration, $\lambda_i = \lambda_i(\tau)$ is a variable across the trajectory $\tau$, as a result, taking the derivative against $\lambda_i(t)$ will be meaningless since the values of the $\lambda_i(\tau)$ along the trajectory is independent of the current value of $\lambda_i$ at time $t$ for all $\tau < t$.
>
> > ... I am not sure about the motivation for the method.
>
> The motivation for our method comes from the finite sum approximation of the integration. The calculation is more intuitive under the form of a finite sum: Since we are calculating the derivative w.r.t. $\lambda_i$ at time $t$, its only contribution to the integral comes from the last term at time $t$. Note that for all terms with $\tau < t$, $\lambda(\tau)$ is independent of $\lambda(t)$. As a result, the only term we get is $-\alpha \dfrac{\partial L_i(\theta, t)}{\partial \theta}$, i.e. the term $ -\alpha \dfrac{\partial L_i}{\partial \theta}$ in the main text. We will explain this step in detail and shortly update the manuscript.
>
> **2. Experiments**
>
> > One thing I observed is that the metrics for the NYUv2 run for the main baseline, Auto-lambda, are different in your submission to the original Auto-lambda paper.
>
> The numbers in Table 2 are the results of our run of the experiments, where the setups, including the metrics, are the same as described in the Auto-$\lambda$ paper. The only distinction lies in the notational conventions we follow in Table 2 in our paper and Table 1 in the Auto-$\lambda$ paper. We will clarify this in the upcoming version.
>
> > Can you report actual memory and runtime usage of your runs?
>
> The memory usage is as follows: On the NYUv2 dataset, Equal weight uses 4944 MB memory, Auto-$\lambda$ uses 7948 MB memory, and our method uses uses 5154 MB memory. On the Cityscapes dataset, Equal weight uses 4206 MB memory, Auto-$\lambda$ uses 6002 MB memory, and our method uses 4352 MB memory. This confirms that our method uses much less memory than the state-of-the-art Auto-$\lambda$.
>
> Reporting the actual runtime is more nuanced. On our cluster, every run is assigned with a fixed amount of GPU, but the CPUs are shared across all jobs. The runtime is related the current CPU load, which makes it hard for benchmarking. We nonetheless report the runtimes across all runs as follows: Equal weight takes 0.76s per step, Auto-$\lambda$ takes 0.92s per step, and our method takes 0.67s per step.

---

> > ### Comment · Reviewer_DDnz · 2024-12-03
> > **Response**
> >
> > Thank you for the clarifications, and sorry for my slow response!
> >
> > Re 1. Thank you for your response. However, I could not find an updated derivation of the method in the paper. Are you planning to post a revision? As it is now, according to your response the derivation of the method with integrals seems to be incorrect.
> >
> > Re 2. Thank you for clarifying. Do you have an hypothesis for what could cause the differences in your runs?
> > Regarding memory reports, this information would be a useful addition to the paper in my opinion.

---

> > > ### Author Response · Authors · 2024-12-10
> > >
> > > Thank you for the reply and sorry for our slow response as well !
> > >
> > > **Re 1.** We have already updated the submission of the paper.
> > >
> > > > As it is now, according to your response the derivation of the method with integrals seems to be incorrect.
> > >
> > > This is not true. The derivation with integrals has always been correct in our submission; instead, as explained in [our earlier reply](https://openreview.net/forum?id=bowQ2rPTYW&noteId=KOWzHZGX07), your proposed derivation by interchanging the derivative and the integral is not correct.
> > >
> > > To make things clearer, we have **explicitly** labeled the dependencies of the term $\lambda$ in our revised submission (see page 5). As you can see. the derivative is with respect to $\lambda(t)$ but the term in the intergral is with respect to $\lambda(\tau)$, which makes the derivative and the integral inexchangeable. Furthermore, due to $\lambda(\tau)$ is independent of $\lambda(t)$ for all $\tau<t$, the integral only contributes to $\partial/\partial \lambda(t)$ when $\tau=t$, therefore completing our derivation. To avoid confusion, we have explained our argument below the equations, please refer to the texts colored in red on page 5.
> > >
> > > **Re 2.** Thank you for raising this point. We have included a detailed analysis of both memory and runtime in the appendix (see page 15). Regarding the discrepancies in the results, we hypothesize that they may stem from differences in the Python environment and/or variations in the GPU/CPU used during computation. However, verifying this hypothesis would be quite challenging.

---

> > > > ### Comment · Reviewer_DDnz · 2024-12-16
> > > > **Response to revision**
> > > >
> > > > Dear authors,
> > > >
> > > > thank you for uploading a revision.
> > > >
> > > > I think that the derivation on page 5 is still mathematically incorrect: assume that $\lambda_i(\tau)=\tau$ and that $\partial L_i/\partial \theta_{enc} = 1 $.
> > > >
> > > > Then, I get $h(t):=\int_0^t \lambda_i(\tau) d\tau = (1/2)t^2$. But then
> > > > $$ \partial h(t)/ \partial \lambda_i(t) =  \partial h(t)/ \partial t = t$$
> > > >
> > > > But according to page (5), the result should have been $1$; so something in the derivation must be wrong.
> > > >
> > > > (for this calculation I didn't switch any integral with derivatives)
> > > >
> > > >
> > > > **Regarding the discrepancies:** I understand that tracing this back might be very hard, but to explain these discrepancies purely with different hardware/software setups is unsatisfying. The discrepancy actually changes the conclusion on which method is better for this specific experiment. If there is randomness in this experiment, one solution could be to do multiple runs and report standard deviations.

---

### Review · Reviewer_85T3 · 2024-10-05

**Summary Of Contributions:**

This work studies auxiliary task learning, in which several auxiliary tasks are introduced during training with the goal of improving the performance of the main task. The work propose a framework called AuxiLight, which dynamically update the auxiliary weights during training, and its computational cost and memory cost are lower than Auto-$\lambda$. The derivation of AuxiLight is based on a simple gradient flow analysis, by formulating the gradient flow of the auxiliary weights wrt the main task loss function. The authors also provide a case study of 6D Pose Estimation in which the auxiliary annotations can be obtained for free. The empirical results show that AuxiLight achieves similiar or slightly better performance than Auto-$\lambda$ while significantly reduce the training overhead.

**Audience:**

Yes

**Claims And Evidence:**

Yes

**Requested Changes:**

Please compare with the dynamic weight tuning strategies in standard multi task learning such as "Debabrata Mahapatra and Vaibhav Rajan. Multi-task learning with user preferences: Gradient descent with controlled ascent in pareto optimization. In International Conference on Machine Learning, pp. 6597–6607, 2020.", and discuss whether it's possible to directly extend their method to auxiliary task learning

**Strengths And Weaknesses:**

Strengths:
- This paper clearly demonstrate the background of the problem, and the design of AuxiLight is well motivated.
- The proposed strategies (i.e., AuxiLight and Annotation free case study) are practical and shown to be empirically beneficial.

Weaknesses:
- In my understanding, auxiliary task learning is a special case of multi task learning. Thus, the strategies designed in multi task learning could be adapted to the cases of auxiliary task learning. For example, we may use the dynamic weight tuning strategy in "Debabrata Mahapatra and Vaibhav Rajan. Multi-task learning with user preferences: Gradient descent with controlled ascent in pareto optimization. In International Conference on Machine Learning, pp. 6597–6607, 2020." in auxiliary task learning, by assigning a high user preference to the main task. Could the authors also compare with the dynamic weight tuning strategies in standard multi task learning, and discuss whether it's possible to directly extend their method to auxiliary task learning?
- Is it possible to direcly analyze the discrete case of the gradient flow? what could be the difficulty?

---

> ### Author Response · Authors · 2024-11-18
>
> We than reviewer 85T3 for the positive feedback, especially for acknowledging the contribution of our proposed AuxiLight algorithm. Here are the detailed reply to the reviewer:
>
> **1. Comparison with multi-task learning**
>
> > For example, we may use the dynamic weight tuning strategy in "Debabrata Mahapatra and Vaibhav Rajan. Multi-task learning with user preferences: Gradient descent with controlled ascent in pareto optimization. In International Conference on Machine Learning, pp. 6597–6607, 2020." in auxiliary task learning, by assigning a high user preference to the main task.
>
> Thanks for bringing up the relevant paper [1]. In fact, this paper belongs to a line of work known \textit{Pareto multi-task learning} [1,2], where a vector-valued loss is optimized instead of a single-valued loss. This is explicitly explained in Chapter 3.2 in [1], or in Chapter 3 in [3], and it is a significant difference between auxiliary-task learning and Pareto multi-task learning. Moreover, [1] implements a method where a linear programming problem has to be solved, which increases the time / space complexities and contradicts our goal to minimize such complexities.
>
> We will nonetheless discuss this work and this difference in the manuscript.
>
> **2. Method.**
>
> > Is it possible to directly analyze the discrete case of the gradient flow?
>
> Yes, it is possible. In fact, the intuition of our formulation in Section 3.2 comes exactly from an approximation of the discrete gradient descent step. However, it is more complex to strictly formalize idea with the discrete version due to lack of mathematical tools (the chain rule of derivative for example).
>
> **References**
>
> [1] Mahapatra, Debabrata, and Vaibhav Rajan. "Multi-task learning with user preferences: Gradient descent with controlled ascent in pareto optimization." International Conference on Machine Learning. PMLR, 2020.
>
> [2] Lin, Xi, et al. "Pareto multi-task learning." Advances in neural information processing systems 32 (2019).
>
> [3] Liu, Shikun, et al. "Auto-lambda: Disentangling dynamic task relationships." arXiv preprint arXiv:2202.03091 (2022).

---

### Review · Reviewer_Nbmk · 2024-11-03

**Summary Of Contributions:**

This paper presents two main contributions. First, it introduces a novel learning strategy for handling auxiliary losses, using a technique that adjusts the auxiliary loss coefficients dynamically. The approach leverages a concept similar to the Leibniz integral rule to compute the derivative of these coefficients. Second, it empirically evaluates this learning strategy in the context of 6D object pose estimation.

**Audience:**

Yes

**Broader Impact Concerns:**

No ethical concerns.

**Claims And Evidence:**

No

**Requested Changes:**

See the weakness, clarification, and suggestions above.  In particular, please include the suggested analysis and ablations, which I copy here:

- **Lack of thorough analysis and ablation studies**:
    - Given the dynamic adjustment of auxiliary coefficients, it would be helpful to see visualizations of how these coefficients evolve over the course of training in different experiments.
    - In experiments involving multiple auxiliary tasks, an analysis of the most beneficial auxiliary tasks would strengthen the results. Additionally, an experiment introducing a “useless” auxiliary task would help verify that the method correctly minimizes such coefficients, further validating its effectiveness.

Please provide these analyses, as they can make your contribution more convincing and clear (and I'm open to changing my rating after the response).

**Strengths And Weaknesses:**

## Strengths
- The derivation of the core technique is clear and intuitive, and the ability to automatically adjust auxiliary loss coefficients is a valuable contribution.
- The empirical results presented in the evaluation tables are competitive with existing approaches, often outperforming them.

## Weaknesses
- **Unclear connection to the application task (6D pose estimation)**:
    - The link between the proposed learning strategy and the chosen task (6D pose estimation) is not well-explained. It is unclear why 6D pose estimation was selected over more common ML tasks like object detection or segmentation.
    - This lack of clarity impacts the Abstract and Introduction, making it difficult to understand the paper's focus after reading the first few sections.
- **Lack of thorough analysis and ablation studies**:
    - Given the dynamic adjustment of auxiliary coefficients, it would be helpful to see visualizations of how these coefficients evolve over the course of training in different experiments.
    - In experiments involving multiple auxiliary tasks, an analysis of the most beneficial auxiliary tasks would strengthen the results. Additionally, an experiment introducing a “useless” auxiliary task would help verify that the method correctly minimizes such coefficients, further validating its effectiveness.


## Clarification Questions
- Where does the coefficient $\alpha$ originate in the block of equations before Equation 4?

## Suggestions
- Adding a summary at the end of the introduction that clearly outlines the main task, auxiliary tasks, and AuxiLight’s specific contributions would improve readability and help readers better understand the paper’s focus.

---

> ### Author Response · Authors · 2024-11-18
>
> We thank reviewer Nbmk for the positive feedback regarding the derivation and the results of our proposed AuxiLight method. Here are the detailed reply to the questions:
>
> **1. Choice of the task of 6D pose estimation.**
>
> > Unclear connection to the application task (6D pose estimation): The link between the proposed learning strategy and the chosen task (6D pose estimation) is not well-explained.
>
> We have applied our method to 6D pose estimation because we found it to be an interesting example of a domain that comes with freely-available annotation for auxiliary tasks. This is not the case for other ML tasks, such as object detection and segmentation.
>
> In fact, the main drawback of auxiliary-task learning, and multi-task learning, for real-world applications is the lack of annotated data. By applying auxiliary task learning to  6D object pose estimation, we showcase a concrete use of auxiliary learning for real-world problems that does not induce annotation costs, as described in the abstract. Such contribution is also endorsed by other reviewers, for example, *"... the authors also provide a case study of 6D Pose Estimation in which the auxiliary annotations can be obtained for free"*, [as commented by reviewer 85T3](https://openreview.net/forum?id=bowQ2rPTYW&noteId=9HyjXgd1Iw).
>
> We will clarify this in the updated version.
>
> > Adding a summary at the end of the introduction...
>
> Thanks for the suggestion, we will make a summary of contributions at the end of the introduction to increase readablility.
>
> **2. Experiments.**
>
> > ... it would be helpful to see visualizations of how these coefficients evolve over the course of training... an experiment introducing a “useless” auxiliary task would help verify that the method correctly.
>
> Thanks for the suggestion. We have conducted experiments on the NYUv2 dataset with four tasks. In addition to the three tasks of semantic segmentation, disparity estimation and part segmentation as described in the main text, we additionally included a task where we predict a randomly generated Gaussian noise. This additional task is "useless" and could potentially jeopardize the training. We take semantic segmentation as main task, and visualize the evolution of the weights of the other three tasks. The results are shown [here](https://drive.google.com/file/d/1d9zq8b9WXaXPW0om9aSmeeo54UkrXLt7/view?usp=sharing
> ), and our experiment shows that the auxiliary weight for noise prediction remains minimum compared to other tasks, albeit not having descreased to zero. We will include this discussion in the updated version.
>
> **3. Coeefficient in Eq. 4.**
>
> > Where does the coefficient originate in the block of equations before Equation 4?
>
> $\alpha$ is the learning rate used for SGD or gradient flow, as defined earlier in Equation 2.

---

### Decision · Action_Editor_Ra8F · 2025-01-10

**Recommendation:** Reject

**Comment:**

The paper conducts research on how to make auxiliary task learning beneficial to the learning of main task. They propose an algorithm called AuxiLight that adaptively balances losses based on an analysis of training dynamics, and apply the proposed method to a showcase application of 6D pose estimation.

Reviewer DDnz suggests rejection of the paper, given that there might be missing steps in mathematical presentations. Although Reviewers Nbmk and 85T3 lean to accept the paper, they point out that the paper results are less convincing and extensive. From the AE's perspective, the authors claim that the proposed method is a generic auxiliary learning algorithm; however, the single showcase of 6D pose estimation and the less extensive experiments even on this single application cannot well justify the claim and main contributions of the work. AE suggests that the authors may choose to apply the proposed method to more applications with extensive empirical justifications, and may choose to submit the paper again.

**Audience:**

yes

**Claims And Evidence:**

Claims are not justified convincingly.

**Resubmission Of Major Revision:**

The authors may consider submitting a major revision at a later time.